# Cognitive Fatigue in Autoregressive Transformers: Formalization and Measurement

**Riju Marwah** [1 2]  **Ritvik Garimella** [2]  **Vishal Pallagani** [2]  **Atishay Jain** [2 3]  **Michael Stewart** [2]  **Amit Sheth** [2 4]

## Abstract

Autoregressive language models frequently degrade during long-horizon generation, producing repetitive text, losing instruction adherence, and exhibiting unstable entropy. Despite the prevalence of these failures, practitioners lack online diagnostics to detect them in real-time as they occur. We formalize this degradation as *cognitive fatigue*, a measurable generation-time state characterized by decay in attention to the original prompt, representational drift, and entropy miscalibration. We introduce the **Fatigue Index (FI)**, a lightweight, model-agnostic diagnostic that aggregates these three signals under explicit axioms (monotonicity, boundedness, interpretability) enabling reliable runtime monitoring. Across nine models (1B–13B parameters), FI trajectories exhibit structured temporal dynamics, predict task degradation (AUROC = 0.95) and repetition ($\rho = 0.94$), and reveal non-monotonic scaling behavior: instruction-tuned models below 3B exhibit faster collapse than base models, with this trend reversing at 7B. Stress analyses further show that FI onset accelerates under longer contexts, middle-positioned evidence, and reduced numerical precision. These results establish cognitive fatigue as a coherent and measurable phenomenon, and position FI as a principled tool for runtime reliability monitoring in production LLM systems.

## 1. Introduction

Large Language Models (LLMs) perform well on short prompts, but systematically degrade during long generations and context-heavy workflows (Liu et al., 2023). Practitioners observe loss of instruction adherence, growing repetition, and unstable token entropy, often without a realtime signal that a generation is deteriorating (Holtzman et al., 2020). This lack of runtime transparency is particularly problematic in long-horizon production deployments such as multi-step reasoning, tool use, and dialogue, where small deviations accumulate into factual errors or unsafe outputs(Dziri et al., 2023). Existing mitigations largely operate at training time or via offline evaluation and do not indicate when a specific generation becomes unreliable (Su et al., 2021).

We study this phenomenon as **cognitive fatigue** in LLMs: a progressive, within-run degradation in instruction adherence, representation stability, and predictive calibration. We operationalize fatigue using three lightweight inference-time signals: decay in attention to the prompt, embedding drift, and deviation of next-token entropy from a healthy range. Each signal corresponds to an interpretable failure mode and requires no retraining or modification of model weights.

To make these signals actionable, we introduce the **Fatigue Index** (FI), a normalized and interpretable aggregation grounded in explicit **axioms** that ensure stability and comparability for online monitoring. We take a foundational stance, asking whether cognitive fatigue can be rigorously defined and reliably measured. We **empirically validate FI** through token-level trajectories, predictive alignment with task accuracy and error proxies, and scaling analyses under long-context (varying input length) stress for models ranging from 1B to 13B. We focus on models up to 13B to balance diagnostic sensitivity with experimental tractability across diverse architectures.

**Contributions.**

1. A formalization of cognitive fatigue for nine decoder-only models grounded in three token-level, model-agnostic signals measurable at inference time.
2. The Fatigue Index (FI) with axioms & properties that justify its form and enable interpretable, stable monitoring.
3. Empirical validation that FI predicts quality-relevant outcomes and is robust across prompts, seeds, lengths, and model sizes; and a hysteresis-based alerting scheme with fewer false triggers.

[1]Guru Gobind Singh Indraprastha University, India [2]Artificial Intelligence Institute, University of South Carolina, USA [3]Indian Institute of Technology, Kanpur, India [4]Indian AI Research Organization, India. Correspondence to: Riju Marwah <marwah.riju@gmail.com>.

*Proceedings of the 43rd International Conference on Machine Learning*, Seoul, South Korea. PMLR 306, 2026. Copyright 2026 by the author(s).

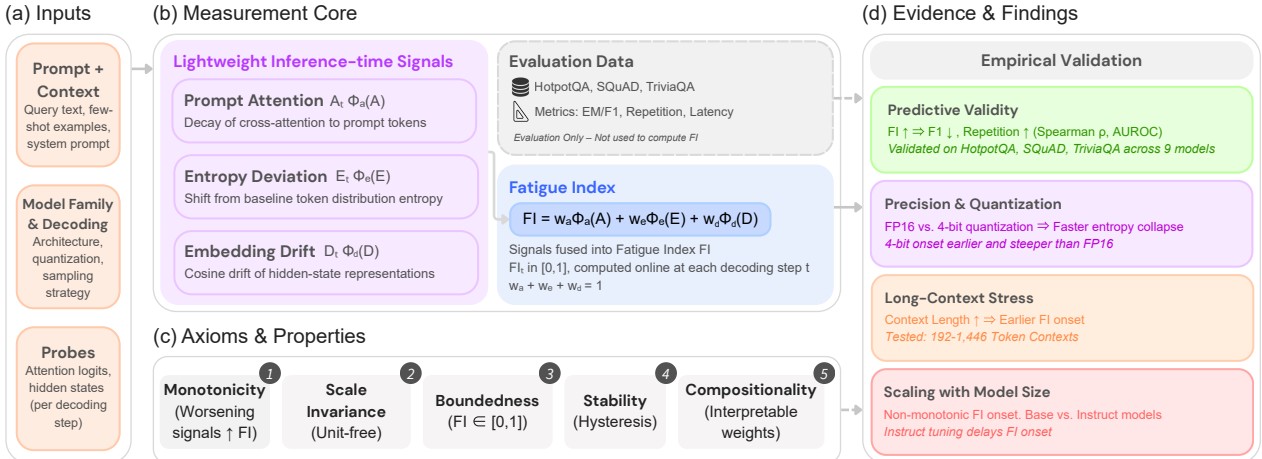

*Figure 1.* **Overview of the cognitive fatigue measurement framework. (a)** A prompt and context are fed to a decoder-only model, with probes extracting attention logits and hidden states. **(b)** Three signals are computed online at each decoding step: prompt-attention decay, entropy deviation, and embedding drift, normalized and aggregated into the Fatigue Index (FI). Evaluation datasets are used only to validate FI, not to compute it. **(c)** FI is grounded in five axioms: monotonicity, scale invariance, boundedness, stability, and compositionality. **(d)** FI is validated against repetition and F1 drop, precision stress (FP16 vs. 4-bit), long-context (varying input length) stress, and scaling behavior across nine models from 1B to 13B parameters.

## 2. Related Work

Liu et al. (2023) show that language models exhibit systematic performance degradation when relevant information appears in the middle of long contexts, characterizing positional bias at the task level, whereas we introduce token-level diagnostics that are measurable online during generation. Su et al. (2021) introduce RoPE, which encodes positional information through rotation matrices, and Peng et al. (2023) build on this idea by proposing YaRN, which combines NTK-by-parts interpolation with attention temperature scaling to extend effective context windows. In contrast, we identify the naturally decaying inter-token dependencies induced by RoPE as a contributing factor to attention dilution and increased stress under long-context conditions. Zhang et al. (2025) propose Recursive Language Models (RLMs), which treat prompts as external environments and rely on recursive sub-calls to process contexts beyond fixed window limits, while our work focuses on measuring the internal degradation that arises during standard autoregressive decoding. Holtzman et al. (2020) characterize neural text degeneration and propose nucleus sampling, and Welleck et al. (2020) introduce unlikelihood training to reduce repetition, whereas we operationalize degeneration as an online state that can be monitored through entropy and distributional drift without modifying training or decoding. Braverman et al. (2020) demonstrate that language models are often miscalibrated and that their entropy rates drift upward over time, and Zhang et al. (2024) and Li et al. (2024b) propose entropy-based dynamic temperature adjustment strategies for decoding, while we treat deviations from

a healthy entropy band as one component of a broader notion of model fatigue rather than as a control signal. Farquhar et al. (2024) introduce semantic entropy for hallucination detection, and surveys by Anh-Hoang et al. (2025) and Alansari & Luqman (2025) categorize detection methods into token-level uncertainty estimation and self-consistency based approaches, whereas our framework enables continuous monitoring during generation by combining entropy with attention-based and drift-based signals. Joglekar et al. (2025) train models to self-report policy violations through explicit confessions, which rely on eliciting introspective behaviors, while we directly measure internal dynamics from attention patterns, hidden representations, and output distributions. Li et al. (2023) introduce Inference-Time Intervention (ITI) to improve factuality by shifting activations along truth-correlated directions, and subsequent work extends this paradigm to cognitive reasoning steering (CREST) (Kadem & Zheng, 2026) and to prototype-based dynamic steering (Kayan & Zhang, 2025), whereas our contribution is purely diagnostic and provides guidance on when and where such interventions may be most beneficial. Overall, we uniquely aggregate three complementary signals into a single axiomatic index that is measurable online and requires no retraining, thereby unifying previously isolated symptoms of long-horizon degradation into a coherent and interpretable diagnostic.

---

**Research Question 1**

*Can cognitive fatigue be formally defined and reliably measured?*

---

# 3. Cognitive Fatigue: Definition & Signals

We define cognitive fatigue in large language models as the gradual degradation of coherence, calibration, and reliability during extended autoregressive decoding. Unlike isolated errors or random noise, fatigue is systematic: the model progressively loses focus on the original instructions, its internal hidden states drift from their initialization, and its output distribution collapses toward overconfident, repetitive continuations. Importantly, this phenomenon is not rare or adversarial, it arises naturally from autoregressive transformer decoders at long horizons.

We formalize this fatigue as a runtime risk indicator measured at every decoding step, rather than a subjective impression. We operationalize fatigue through three lightweight, token-level signals, each capturing a different facet of degradation. Together, these signals form the foundation of the fatigue index, enabling runtime monitoring and intervention.

## 3.1. Signals

We define three complementary signals, each directly observable at inference time and corresponding to a distinct failure mode. All signals are computed per token.

**Attention-to-prompt (attention decay).** A primary failure mode in long-horizon generation is **loss of instruction adherence**: as decoding progresses, the model may increasingly condition on its own recent outputs rather than the original task specification. Transformer decoders expose this phenomenon through attention weights. As the context grows, attention mass allocated to the initial prompt decay even with relevant instruction. (Li et al., 2024a)

We compute the mean last-layer attention weight from the current token to the tokens in initial prompt slice. A declining trend indicates that the model is increasingly underutilizing its instructions.

**Embedding drift.** Long-horizon decoding repeatedly updates a shared residual stream, allowing small perturbations to accumulate over time. As a result, hidden states gradually drifts away from the representational subspace induced by the prompt, even when the generated text remains locally fluent. This internal drift precedes surface level incoherence and repetition without immediate textual signals. (Li et al., 2024a)

At each step $t$, we measure the Euclidean distance $D_t = |h_t - h_0|_2$, where $h_t$ is the hidden state at step $t$ and $h_0$ is the hidden state of the last prompt token. Persistent growth in $D_t$ indicates that the model's internal representation is diverging from the task context, capturing a latent degradation process that is not directly observable from surface text.

**Entropy Collapse.** During extended generation, models often become overconfident (low-entropy, repetition, degeneracy), or erratically uncertain (high-entropy). Next-token entropy provides a direct view of this calibration state at inference time. (Holtzman et al., 2020)

We compute the Shannon entropy of the next-token softmax distribution. Entropy within a calibrated range indicates stable generation; persistent low values indicate over-confidence and repetitive degeneration. In contrast, abnormally high entropy signals indecision.

## 3.2. Fatigue Index: Definition, Normalization, and Weighting

Individual signals are insufficient: attention-to-prompt can drop momentarily without failure, embedding drift grows slowly even in stable runs, and entropy fluctuates with content. A usable diagnostic must therefore integrate these signals into a single interpretable and comparable metric.

We require a real-time control signal imposing constraints on complexity: the aggregator must be computationally cheap, transparent, and robust to perturbations. We adopt a linear aggregator over normalized signals, prioritizing interpretability and inspectability over statistical optimality.

Formally, we defined the Fatigue Index at decoding step $t$ as

$$FI_t = w_A\, \phi_A(A_t) + w_D\, \phi_D(D_t) + w_E\, \phi_E(E_t), \quad (1)$$

where $\phi_A, \phi_E, \phi_D \in [0,1]$ are monotone normalization maps and $D_t$, $E_t$ are the raw signal values, and $w_A + w_D + w_E = 1$. Higher values indicate greater fatigue.

The normalization maps convert heterogeneous raw signals into a consistent, unit-free scale, enabling meaningful aggregation. We adopt theory-guided fixed weights $w_A = 0.40, w_E = 0.35, w_D = 0.25$, empirically validated in Section 7. The ordering $w_A \geq w_E \geq w_D$ encodes domain priors: (i) $w_A$: prompt attention most directly governs instruction following (Qin et al., 2024); (ii) $w_E$: entropy governs degeneracy and repetition (Holtzman et al., 2020); (iii) $w_D$: drift signals longer-horizon instability but is noisier (Li et al., 2024a). With frozen discretizations of these weights, all experiments are executed for comparability and to avoid adaptive tuning. Weight ordering reflects the intended diagnostic role of each signal, not its standalone power as a degeneration detector; weight sensitivity analysis and learned-weight comparisons remain open extensions. For long-horizon degradation, FI is interpreted as an instantaneous risk score. Once calibration is fixed, FI admits a constant scale across runs and models since each term is normalized and bounded. Crucially, FI reflects the internal reliability state of the generation process. More complex aggregators would conceal attribution and complicate online use. In contrast, FI enables attribution by exposing per-signal contributions and supports simple stabilizing mechanisms such as hysteresis, reducing false alerts.

# 4. Axioms of the Fatigue Index

We specify axioms that any admissible online fatigue measure should satisfy. These axioms are normative design requirements, they define what a well-behaved runtime diagnostic should do, and motivate the structure of the Fatigue Index accordingly. They are not empirically verified laws of model internals, nor do they constitute a mechanistic theory of why degradation occurs; rather, they constrain the space of valid online fatigue measures to those with interpretable, stable, and directionally correct behavior.

Let $A_t$ denote prompt-directed attention, $E_t$ output entropy, and $D_t$ representation drift at decoding step $t$. Let $E^\star$ denote the target entropy (calibrated range center). Let $F_t = F(A_t, E_t, D_t)$ be any candidate fatigue score computed online from these signals.

**A1 (Monotonicity).** Fatigue must respond directionally to worsening reliability signals:

$$A'_t \leq A_t \;\Rightarrow\; F(A'_t, E_t, D_t) \geq F(A_t, E_t, D_t),$$

$$|E'_t - E^\star| \geq |E_t - E^\star| \;\Rightarrow\; F(A_t, E'_t, D_t) \geq F(A_t, E_t, D_t),$$

$$D'_t \geq D_t \;\Rightarrow\; F(A_t, E_t, D'_t) \geq F(A_t, E_t, D_t).$$

Thus, reduced attention, increased entropy deviation, or increased representational drift must monotonically increase fatigue.

**A2 (Scale Invariance).** Monotone reparameterizations of the raw signals must preserve fatigue ordering:

$$\mathrm{rank}(F(\phi_A(A), \phi_E(E), \phi_D(D))) = \mathrm{rank}(F(A, E, D)),$$

for any strictly monotone maps $\phi_A, \phi_E, \phi_D$. This ensures that fatigue ordering is preserved under monotone re-parameterizations of the raw signals, it does not imply that absolute FI values are invariant across decoding regimes. In the implemented metric (Section 6.2), fixed normalization maps choose one calibrated representation for boundedness and thresholdability; scale invariance holds in the ordinal sense, not as universal comparability of absolute values.

**A3 (Boundedness).** Fatigue lies on a fixed, interpretable scale: $F_t \in [0, 1]$. This supports stable thresholds and consistent online monitoring.

**A4 (Temporal Stability).** Small, transient perturbations in the underlying signals should not induce large, instantaneous changes in fatigue:

$$\|F_t - F_{t-1}\| \leq L \, \|(A_t, E_t, D_t) - (A_{t-1}, E_{t-1}, D_{t-1})\|,$$

for some $L > 0$. This property prevents spurious oscillations under inference-time noise.

**A5 (Compositionality).** Fatigue should decompose into interpretable per-signal contributions:

$$F_t = \sum_k w_k \, g_k(s_{k,t}), \qquad w_k \geq 0,$$

where $s_{k,t} \in \{A_t, E_t, D_t\}$ and $g_k$ are monotone maps. This enables attribution and controllable weighting of failure modes.

Together, these axioms require directional correctness, scale invariance, bounded range, temporal stability, and interpretability. They constrain the space of valid online fatigue measures without uniquely specifying a construction; the Fatigue Index defined in Section 3.2 is one such realization. Moreover, if a fatigue measure is additive and satisfies A1–A3 and A5, it admits a representation as a weighted sum of bounded, monotone signal transforms (Theorem E.1, Appendix E).

> **Research Question 2**
>
> *Which architectural and numerical factors are associated with the emergence of cognitive fatigue?*

# 5. Mechanisms of Cognitive Fatigue

### 5.1. Where Fatigue Creeps In

Cognitive fatigue does not arise from a single failure mode, but from interacting architectural and numerical pressures that accumulate during long-horizon decoding. Table 1 summarizes the primary mechanisms, the signals they affect, and the empirical evidence supporting each.

### 5.2. Architectural Stress Analyses (Empirical Probes)

To identify architectural pressures associated with fatigue, we perform controlled ablations varying context length, positional bias, and numerical precision.

**Context-length stress.** We vary neutral filler length while holding question and evidence constant, thereby isolating sequence length effects independent of task difficulty. As show in Figure 2, longer contexts are associated with earlier and more sustained collapse of prompt-directed attention. Embedding drift increases in all conditions but becomes more variable as length grows, while entropy exhibits larger fluctuations. These trends are consistent with attention dilution and the accumulation of residual deviations under extended sequences.

**Positional Sensitivity.** We place identical evidence at front, middle and end positions with constant length. Figure 3 shows front-positioned evidence receives higher attention farthest in token distance to the decoding head. This positional bias explains systematic under-utilization of late context, accelerating prompt-forgetting.

| Mechanism | Description | Observed Effect | Evidence / Citation |
|---|---|---|---|
| **Attention dilution** (architectural pressure) | As sequence length grows, softmax attention mass over early prompt tokens is diluted; positional biases (RoPE, ALiBi) further favor recent tokens. | Declining attention to prompt/evidence; worse F1 when evidence is distant from the decode head. | Positional ablation (front/middle/end); (Vaswani et al., 2017; Press et al., 2022; Su et al., 2021) |
| **Residual drift** (state accumulation) | Errors compound in the shared residual stream; deviations propagate rather than cancel due to autoregressive updates and LayerNorm dynamics. | Monotonic increase in embedding drift $D_t$; higher drift aligns with repetition and lower F1. | Drift trajectories; predictive-validity analysis; (He et al., 2016; Ba et al., 2016; Xiong et al., 2020) |
| **Entropy collapse** | Autoregressive training favors sharp predictions; greedy or low-temperature decoding amplifies overconfidence, especially under quantization. | Entropy leaves a healthy band; repetitive and degenerate outputs late in generation. | Entropy trajectories; precision ablation; (Bengio et al., 2015; Huszár, 2016; Dettmers et al., 2023; Lin et al., 2023) |
| **Context length stress** | Long contexts strain numerical precision in KV caches; RoPE phase saturation without scaling degrades long-range recall. | Biased attention scores and unstable output distributions as length increases. | Context-length stress tests; (Press et al., 2022; Chen et al., 2023; Peng et al., 2023) |
| **Decoding hyperparameters** | Temperature, top-$p$, repetition penalties reshape entropy and attention dynamics; extreme settings can accelerate collapse. | Earlier entropy collapse or repetition under long horizons. | Controlled decoding (fixed $top\text{-}p = 0.95$); (Holtzman et al., 2020; Welleck et al., 2020; Li et al., 2016) |

*Table 1.* Architectural and numerical mechanisms associated with cognitive fatigue in decoder-only language models, the signals they affect, and empirical evidence linking them to long-horizon degradation.

**Precision and quantization.** For isolation of numerical effects, we compare FP16 and 4-bit NF4 decoding under matched prompts and seeds. Attention and drift trajectories remain similar, but entropy exhibits deeper, more variable collapse under quantization (Figure 4). Reduced precision destabilizes predictive calibration rather than disrupting prompt focus or representation stability, increasing susceptibility to degeneration. Supplementary precision stress results on HotpotQA and TriviaQA are reported in Appendix D (Figure 9 and 10).

These probes show fatigue arises from interacting architectural pressures: longer contexts dilute attention and amplify drift, positional bias suppresses late evidence, and reduced precision destabilizes entropy. These mechanisms explain signal trajectories in Section 7 and motivate FI as a composite diagnostic.

## 6. Estimation of the Fatigue Index

We specify the inference-time computation of the Fatigue Index (FI) to enable reproducible implementation. This section defines the per-token signals, their normalization and aggregation, and the calibration procedure.

### 6.1. Per-token Signals

Let $t$ index generated tokens. All quantities are computed online with constant-time overhead.

**Prompt attention** ($A_t$). Mean last-layer attention mass allocated to a fixed prompt slice, reflecting continued conditioning on the instruction:

$$A_t = \frac{1}{H} \sum_{h=1}^{H} \sum_{j=1}^{\min(K, L_p)} \text{Attn}_h(x_t, x_j), \qquad (2)$$

with $L_p$ as prompt length and $K$ as prompt slice size.

**Output entropy** ($E_t$). Shannon entropy of $P(x_{t+1} \mid x_{\leq t})$ computed from model logits, serving as a proxy for predictive calibration.

**Embedding drift** ($D_t$). Euclidean distance from the final prompt state:

$$D_t = \|h_t - h_0\|_2, \qquad (3)$$

where $h_0$ is the top-layer hidden state of the final prompt token.

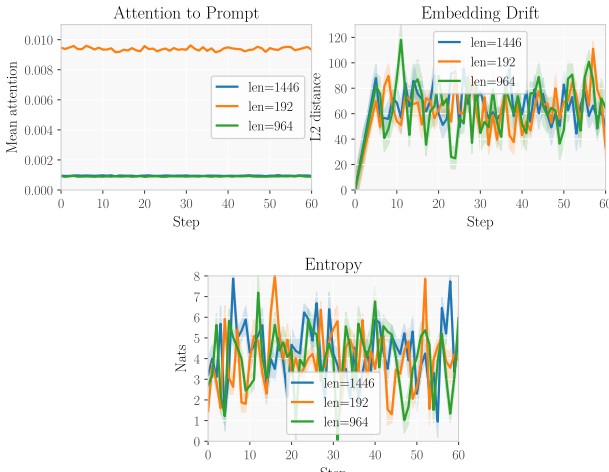

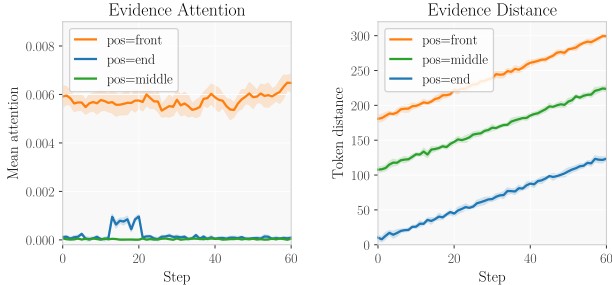

*Figure 3.* Attention to evidence tokens as a function of their position in the prompt, measured over 60 generation steps. Evidence placed at the front of the context receives 5–10× higher mean attention than evidence placed in the middle or at the end, and this gap is stable across all steps, confirming a strong primacy bias that may cause the model to underweight late-context evidence during reasoning.

### 6.3. One-time Calibration and Implementation

Calibration hyperparameters (entropy band, drift cap, and weights) are fixed from a small preliminary pass and then frozen before evaluation. Fixed default values for all hyperparameters are reported in Appendix A (Table 6). Last-layer attentions are obtained using eager attention. All normalization maps are stateless and incur negligible overhead.

> **Research Question 3**
>
> *Does Fatigue Index reliably track long-horizon degradation and predict performance loss across models, prompts, and decoding settings?*

## 7. Empirical Validation

We evaluate whether the Fatigue Index (FI) behaves as a meaningful, general, and online indicator of generation-time degradation. Unless otherwise stated, all experiments use OPT-2.7B. We analyze 27,405 generated sequences drawn from three distinct domains: HotpotQA (reasoning) (Yang et al., 2018), TriviaQA (knowledge) (Joshi et al., 2017), and SQuAD (comprehension) (Rajpurkar et al., 2016).

### 7.1. Cross-Dataset Behavior of Fatigue

We examine whether fatigue is specific to a particular benchmark or whether it arises consistently across task domains. Figure 5 shows the mean FI trajectory as a function of generated token position for each dataset. Despite substantial differences in task structure and prompt style, FI exhibits a highly consistent accumulation pattern across all three domains. In each case, FI rises smoothly with generation length and approaches a similar asymptotic range.

Table 2 reports the mean FI and repetition rates for each dataset. Mean FI values are tightly clustered around ≈

*Figure 2.* Effect of context length on attention, drift, and entropy trajectories across 60 steps. Longer contexts (len=1446) exhibit near-zero prompt attention throughout, indicating early attention collapse, while shorter contexts (len=192) maintain substantially higher attention. Embedding drift and entropy both show increased variability with context length, suggesting that longer inputs destabilize internal representations, which is a key failure mode to monitor in deployment.

### 6.2. Normalization and Aggregation

Raw signals are mapped to unitless penalties in $[0, 1]$ using fixed monotone transforms:

$$\phi_A(A_t) = 1 - \text{clip}(A_t, 0, 1),$$

$$\phi_E(E_t) = \text{clip}\left(\begin{cases} \dfrac{H_\ell - E_t}{H_\ell} & \text{if } E_t < H_\ell, \\ 0 & \text{if } H_\ell \le E_t \le H_u, \\ \dfrac{E_t - H_u}{\beta} & \text{if } E_t > H_u, \end{cases} 0, 1\right),$$

$$\phi_D(D_t) = \text{clip}\left(\dfrac{D_t}{\kappa}, 0, 1\right),$$

where $\text{clip}(x, a, b) = \min\{\max\{x, a\}, b\}$ and $(H_\ell, H_u, \beta, \kappa)$ are fixed from a small preliminary pass and then frozen for all reported experiments.

Entropy is penalized only when it deviates from a fixed "healthy band," capturing both overconfidence and excessive uncertainty. Drift is scaled by a fixed cap so that transient deviations remain small while persistent divergence saturates the signal.

FI is computed via Eq. 1 with weights $w_A = 0.40$, $w_E = 0.35$, and $w_D = 0.25$. For online alerting, FI is smoothed over a short window and passed through hysteresis with distinct activation and deactivation thresholds, suppressing transient jitter without altering signal definitions.

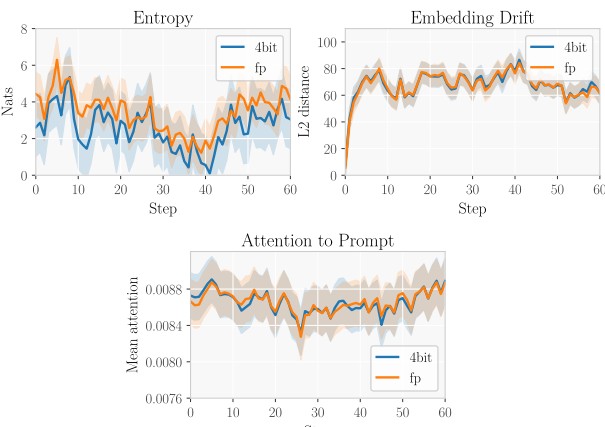

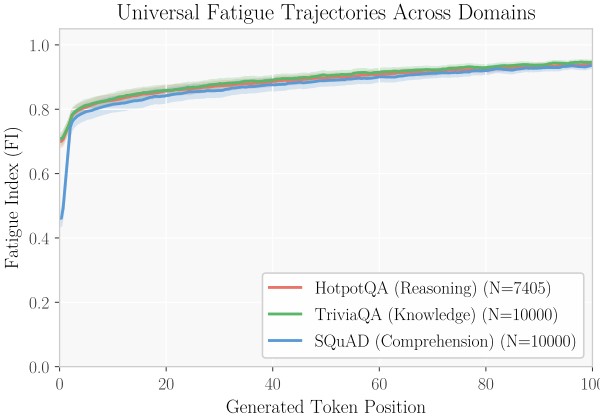

*Figure 4.* Effect of numerical precision (FP16 vs. 4-bit NF4) on entropy, attention, and embedding drift under matched prompts and seeds. Attention to prompt and embedding drift trajectories remain nearly identical across precisions, indicating that quantization does not disrupt prompt focus or representation stability. Entropy, however, exhibits deeper and more variable collapse under 4-bit NF4, suggesting that reduced precision primarily destabilizes predictive calibration and increases susceptibility to degeneration. Note: the 4-bit NF4 entropy trajectory is reconstructed from paper-reported statistics due to a plotting error in the original experimental run; attention and drift curves are extracted directly from experimental output.

*Figure 5.* Universal fatigue trajectories across domains (OPT-2.7B). Mean Fatigue Index (FI) over generated token positions for HotpotQA (reasoning), TriviaQA (knowledge), and SQuAD (comprehension). Shaded regions denote 95% confidence intervals. All three datasets exhibit a sharp FI rise within the first 5 tokens followed by a consistent logarithmic increase toward saturation near FI 0.94, indicating that output fatigue is a domain-agnostic phenomenon rather than a task-specific artifact.

0.82, and higher FI coincides with elevated repetition. This consistency rules out dataset-specific artifacts and supports fatigue as a domain-agnostic degradation process intrinsic to long-horizon decoding.

### 7.2. Predictive Validity

We evaluate whether FI predicts repetition-based degeneration proxies; comprehensive EM/F1 forecasting is beyond the scope of the current analysis. Table 3 reports Spearman correlations between FI and repetition computed over the full generation (up to 120 tokens) and over only the first 20 tokens.

Across all datasets, FI exhibits strong correlation with failure. In contrast, correlations computed from the first 20 tokens are substantially weaker. The low correlation of early tokens ($\rho \approx 0.4$) compared to the full sequence ($\rho > 0.8$) is expected: degradation is a cumulative, non-Markovian process, and FI is designed to track trajectories rather than provide strong single-step early warnings. The full-trajectory correlation is therefore the primary supported claim; early-token correlation is reported to characterize the onset dynamics, not as a standalone predictive result.

These results establish FI as a predictive signal of degeneration rather than a purely descriptive statistic. However, one limitation is that repetition ratio is not fully independent of entropy-based signals, since repetitive outputs tend to

exhibit low entropy. While Figure 5 demonstrates that FI achieves higher AUROC than entropy alone, this only partially addresses the concern regarding their correlation. FI is therefore best understood as a detector of long-sequence text generation, rather than as a comprehensive reliability metric for failures such as hallucinations or instruction drift.

### 7.3. Online Stability

For deployment, an online diagnostic must avoid excessive threshold oscillations. We therefore compare a naive threshold on FI with a hysteresis-based decision rule. Table 4 reports the number of alert flips per generation.

Across all datasets, hysteresis reduces jitter by more than 91%. This demonstrates that FI supports stable online monitoring and is production-ready for real-time control or intervention policies without introducing spurious flickering behavior. Robustness of FI across random seeds and prompt styles is further analyzed in Appendix C1

### 7.4. Benefit of Aggregation

We test whether aggregation is necessary by comparing FI against its individual components (entropy, drift, and attention) in discriminating severe degeneration on HotpotQA. Table 5 reports AUROC values.

On HotpotQA, the aggregated FI achieves an AUROC of 0.978, significantly outperforming the strongest individual baseline (Drift: 0.951). Attention alone performs poorly as a standalone degeneration detector (AUROC = 0.31), yet it is assigned the highest weight because it most directly

*Table 2.* Cross-dataset robustness. Mean FI and repetition rate (fraction of generated tokens that are exact n-gram repeats) across three domains. Note that repetition rate serves as a behavioral failure proxy and does not capture all dimensions of task accuracy.

| Dataset | $N$ | Mean FI | Repetition Rate |
|---|---|---|---|
| HotpotQA (Reasoning) | 7405 | 0.815 | 0.404 |
| SQuAD (Comprehension) | 10000 | 0.812 | 0.423 |
| TriviaQA (Knowledge) | 10000 | 0.833 | 0.467 |

*Table 3.* Predictive validity: Spearman correlation between FI and repetition-based failure proxy (repetition ratio), computed over the full generation (up to 120 tokens) and over only the first 20 tokens. Repetition ratio is used as a failure proxy; comprehensive task accuracy (EM/F1) is reported separately and not captured here.

| Dataset | Correlation (Full FI) | Correlation (First 20) |
|---|---|---|
| HotpotQA (Reasoning) | 0.848 | 0.425 |
| TriviaQA (Knowledge) | 0.820 | 0.404 |
| SQuAD (Comprehension) | 0.856 | 0.375 |

reflects instruction-following behavior rather than surface repetition; Table 5 does not independently validate the specific weight allocation. This justifies the multi-component construction of FI and supports aggregation as necessary for robust detection.

### 7.5. Scaling with Model Size

We further examine how fatigue interacts with model size and instruction tuning across nine models spanning 1B–13B parameters. Results are observational: decoding is controlled, but architectural design, pretraining data, and alignment procedures are not.

In the sub-3B regime, instruction-tuned models exhibit earlier entropy collapse than base models under matched decoding. These scaling observations are explicitly observational: decoding is controlled, but architectural design, pretraining corpora, and alignment procedures vary across families and constitute uncontrolled confounders. This trend reverses near 7B parameters, where instruction-tuned models display improved entropy calibration. Figure 6 illustrates this size–alignment interaction.

At 13B, an outlier behavior is observed: Llama-2-13B-Chat collapses into low-entropy refusal templates despite grammatical output, producing a form of "safety fatigue." This suggests that aggressive alignment can restrict the effective output manifold in ways that mimic cognitive fatigue.

Finally, drift slopes remain approximately constant across model sizes (Figure 7), indicating that all autoregressive models deviate from their initial representations over long horizons. Larger models differ not by drifting less, but by maintaining coherence while drifting. Model-wise drift and entropy statistics supporting these trends are reported in Appendix B (Table 7).

Taken together, these results indicate that fatigue is a general property of long-horizon decoding, modulated by capacity

and alignment but not eliminated by scale.

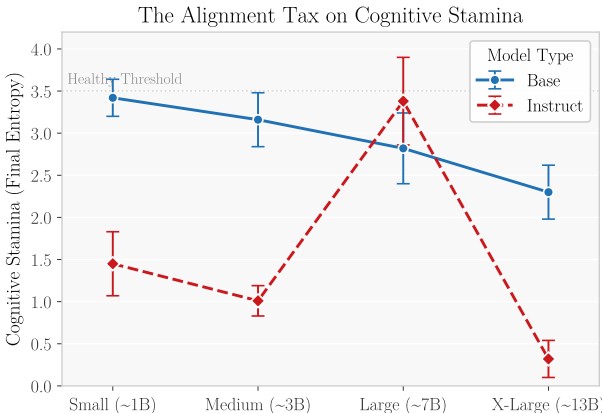

*Figure 6.* Per-family cognitive stamina (mean ± 95% CI) under matched decoding conditions. Base models show a gradual monotonic decline in final entropy with scale, while instruction-tuned models exhibit a non-monotonic pattern, collapsing earlier at sub-3B scales, recovering near 7B, then collapsing sharply at 13B. This divergence suggests that alignment fine-tuning imposes a scale-dependent cost on output diversity that does not diminish with model size. Results are observational; architectural and data confounders remain.

## 8. Conclusion

We formalized cognitive fatigue as a measurable degradation pattern in autoregressive language models and introduced the Fatigue Index (FI), an online, model-agnostic diagnostic aggregating prompt attention decay, representational drift, and entropy miscalibration. Grounded in explicit axioms, FI defines a stable and interpretable scale computable at inference time without retraining or access to training data. Empirically, FI accumulates consistently under long-horizon decoding, generalizes across model sizes and architectural stressors, and anticipates qualitative breakdowns such as rep-

*Table 4.* Online alerting stability under hysteresis. A "flip" is counted each time the alert state toggles between active and inactive within a single generation. Hysteresis uses separate activation and deactivation thresholds ($\theta = 0.50$ and $\theta_{\text{low}} = 0.40$) respectively, reducing alert flips by over 91% across all datasets compared to a naive single-threshold rule. This demonstrates that FI supports stable, production-ready runtime monitoring without spurious oscillation.

| Dataset | Naive Flips/Gen | Hysteresis Flips/Gen | Reduction (%) |
|---|---|---|---|
| HotpotQA (Reasoning) | 18.934 | 1.567 | 91.718 |
| TriviaQA (Knowledge) | 20.889 | 1.433 | 93.140 |
| SQuAD (Comprehension) | 21.085 | 1.674 | 92.060 |

*Table 5.* Benefit of signal aggregation measured by AUROC on HotpotQA. Severe degeneration is defined as sequences in the top quartile of repetition ratio (repetition rate 0.6). The aggregated FI substantially outperforms all individual components, confirming that multi-signal integration is necessary for robust detection.

| Dataset | Fatigue Index (Ours) | Entropy Only | Drift Only | Attention Only (Inv) |
|---|---|---|---|---|
| HotpotQA AUROC | 0.976 | 0.954 | 0.929 | 0.307 |

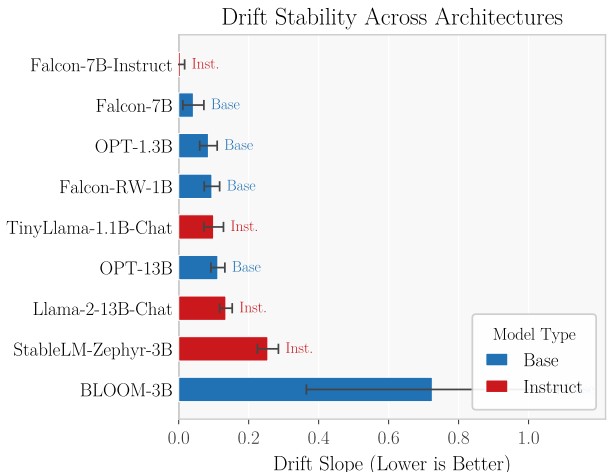

*Figure 7.* Embedding drift slope per model, sorted in ascending order (lower is better). Despite spanning 1B to 13B parameters, most models cluster tightly between drift slopes of 0.08–0.14, with BLOOM-3B as a high-variance outlier. The absence of a clear size-dependent trend confirms that drift stability is largely architecture- and training-regime-dependent rather than a function of scale. Larger models do not drift less, but may drift more coherently.

etition and entropy collapse. These failures are correlated with systematic token-level changes in attention, entropy, and hidden-state drift, enabling degradation to be tracked during decoding rather than only through post hoc task metrics. By surfacing correlated signals of internal instability at runtime, FI reframes long-horizon reliability as an online monitoring problem and supports real-time identification of potentially unreliable segments. FI is designed as a diagnostic that can inform monitoring, logging, or fallback triggers; demonstrating gains from closed-loop intervention is an important open direction not demonstrated in this paper.

**Limitations.** FI requires access to logits, attentions, and hidden states, which may be unavailable through closed APIs. Our evaluation is limited to nine open decoder-only models and QA-style tasks with generations capped at 120 tokens; fatigue dynamics in dialogue, code, and tool use remain unexplored. FI relies on fixed linear aggregation and provides a correlational risk indicator rather than a causal or task-optimal metric, it does not directly measure an internal model mechanism. Absolute FI values depend on decoding and numerical settings and are not directly comparable across arbitrary regimes. The fixed entropy band $[3.8, 5.0]$ and weight vector ($w_A = 0.40, w_E = 0.35, w_D = 0.25$) are calibrated for the evaluated regime; per-model calibration, normalized entropy variants, and learned-weight comparisons are reasonable extensions. Fixed weight transferability across different model families, decoding strategies, and task distributions is unverified. Last-layer attention and L2 drift are lightweight operational probes, not uniquely faithful mechanistic explanations; cosine distance and layer-normalized drift alternatives are worth studying. The primary validated predictive claim is online detection of repetition-based text degeneration; FI does not cover hallucination, factual errors, or instruction-deviation failures not manifesting as repetition. Repetition and entropy signals are partially coupled, and FI may produce false alarms when entropy deviates for task-structural reasons unrelated to degradation. Practitioners would benefit from direct comparison of FI against external runtime monitors such as perplexity-based or semantic-entropy-based diagnostics; such monitor-to-monitor comparison is a limitation of the current work. A key open direction is mechanistic interpretability: identifying circuit-level causes of fatigue and linking FI components to specific internal failure mechanisms. Demonstrating FI-triggered closed-loop intervention gains is also future work.

## Impact Statement

**Intended positive uses.** This work introduced a principled, online diagnostic for monitoring degradation in long-horizon generation in language models. By making degradation observable at runtime, the proposed Fatigue Index (FI) can support more transparent and reliable deployment of language models, for example by activating monitoring, logging, or verification triggers when quality deteriorates. In settings where reliability is essential such as customer support, healthcare, and education, FI may help reduce silent failure by surfacing degradation that would otherwise go unnoticed. Because FI is retrain-free and computer online from inference-time signals, it is compatible with a wide range of open-model deployments and resource-constrained environments.

**Risks and mitigation.** FI is a diagnostic proxy rather than a ground-truth indicator of correctness or safety. Over-reliance of FI as a hard decision could suppress valid but creative outputs or induce misplaced trust. There is also a risk that systems could be optimized to the index without improving underlying task reliability. To mitigate these risks, we recommend using FI as a complementary signal alongside task-level evaluation or human-judgment, and inspecting per-signal components rather than relying on aggregate score.

**Misuse and overinterpretation.** Several misuse risks warrant explicit acknowledgment. First, FI should not be interpreted as a general safety indicator: elevated FI reflects internal signal degradation under the specific conditions studied and does not imply that a generation is harmful, factually incorrect, or unsafe. Second, FI will produce false positives and false negatives in deployment. High FI does not guarantee output failure, and low FI does not guarantee reliable output; threshold-based interventions should be calibrated per application and validated against task-level metrics before deployment. Third, the term "cognitive fatigue" is an intentional analogy chosen for interpretability, not a claim about cognition or experience in language models. Treating this framing literally risks anthropomorphizing model behavior in ways that could mislead users or inflate perceived capabilities. We encourage practitioners to engage with FI as a statistical signal over internal model states, not as evidence of model awareness or intent.

**Societal considerations.** By exposing uncertainty and degradation during generation, FI may counter the perception of unwarranted certainty often conveyed by fluent model outputs, supporting more informed user interaction. The signals used by FI are derived solely from internal model states during decoding and do not require additional user data, preserving user privacy. FI should not be viewed as a substitute for improved training data, model design, or evaluation practices; rather, it is a lightweight diagnostic that complements existing approaches by making failures more measurable and explicit.

Overall, this work advances the measurement and monitoring of long-horizon behavior in language models. While it does not eliminate failure modes, it provides a transparent and model-agnostic tool for studying, comparing, and responsibly deploying autoregressive systems.

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

## A. Fixed Defaults and Calibration

*Table 6.* Fixed defaults used in all Fatigue Index (FI) experiments. Values were selected once from a small preliminary pass and then frozen; no per-item or per-dataset tuning was performed.

| Parameter | Value |
|---|---|
| Prompt slice $K$ | 64 |
| Entropy band $[H_\ell, H_u]$ | [3.8, 5.0] |
| FI smoothing window $L$ | 5 |
| Hysteresis $(\theta, \theta_{low})$ | (0.50, 0.40) |
| Weights $(w_A, w_E, w_D)$ | (0.40, 0.35, 0.25) |
| Probe frequency | every 2 tokens |
| Max context / Max new tokens | 2048 / 120 |
| Decoding (top-$p$, $T$, top-$k$) | 0.95, 1.0, 0 |
| Seeds | {123, 2027} |

## B. Model-wise Summary Statistics for Scaling Experiments

*Table 7.* Scaling behavior across model families. Drift and entropy statistics are computed over long-horizon decoding runs under matched decoding conditions.

| Model | Type | Size (B) | Drift Mean | Drift Std | Entropy Mean | Entropy Std |
|---|---|---|---|---|---|---|
| tiiuae/falcon-rw-1b | Base | 1.0 | 0.102 | 0.069 | 3.587 | 1.780 |
| TinyLlama/TinyLlama-1.1B-Chat-v1.0 | Instruct | 1.1 | 0.104 | 0.062 | 1.430 | 1.779 |
| facebook/opt-1.3b | Base | 1.3 | 0.087 | 0.046 | 3.242 | 1.572 |
| bigscience/bloom-3b | Base | 3.0 | 0.724 | 0.796 | 3.156 | 1.393 |
| stabilityai/stablelm-zephyr-3b | Instruct | 3.0 | 0.250 | 0.078 | 1.004 | 0.861 |
| tiiuae/falcon-7b-instruct | Instruct | 7.0 | -0.001 | 0.096 | 3.375 | 2.470 |
| tiiuae/falcon-7b | Base | 7.0 | 0.041 | 0.080 | 2.826 | 1.885 |
| NousResearch/Llama-2-13b-chat-hf | Instruct | 13.0 | 0.135 | 0.038 | 0.306 | 0.814 |
| facebook/opt-13b | Base | 13.0 | 0.114 | 0.043 | 2.304 | 1.697 |

## C. Robustness of the Fatigue Index

This appendix reports supplementary analyses assessing the stability of the Fatigue Index (FI) under minor experimental variation. These experiments test whether FI behaves as a reproducible internal state rather than as a brittle heuristic tied to specific seeds or prompt formulations.

### C.1. Robustness Across Seeds and Prompt Styles

Using the same model and decoding configuration as in the main experiments, we repeat generation across multiple random seeds and two prompt styles (zero-shot and short chain-of-thought). For each condition, we report the mean FI aggregated over sequences.

Table 8 shows that FI varies minimally across seeds and preserves consistent ordering across prompt styles, indicating that the fatigue signal is not driven by incidental randomness or prompt phrasing.

*Table 8.* Mean Fatigue Index (FI) across random seeds and prompt styles (zero-shot vs. short chain-of-thought).

| Seed | Prompt Style | Mean FI |
|---|---|---|
| 42 | Standard (Zero-shot) | 0.854 |
| 42 | CoT | 0.858 |
| 123 | Standard (Zero-shot) | 0.854 |
| 123 | CoT | 0.858 |

### C.2. Smoothness under Controlled Perturbations

We visualize FI under small controlled perturbations (random seed and prompt style) to assess whether it varies smoothly or exhibits erratic behavior. Figure 8 shows a structured, continuous surface rather than discontinuous or noisy responses.

Regions of elevated FI coincide with regimes associated with downstream degeneration in the main experiments, supporting the interpretation of FI as a cumulative degradation state rather than a brittle heuristic.

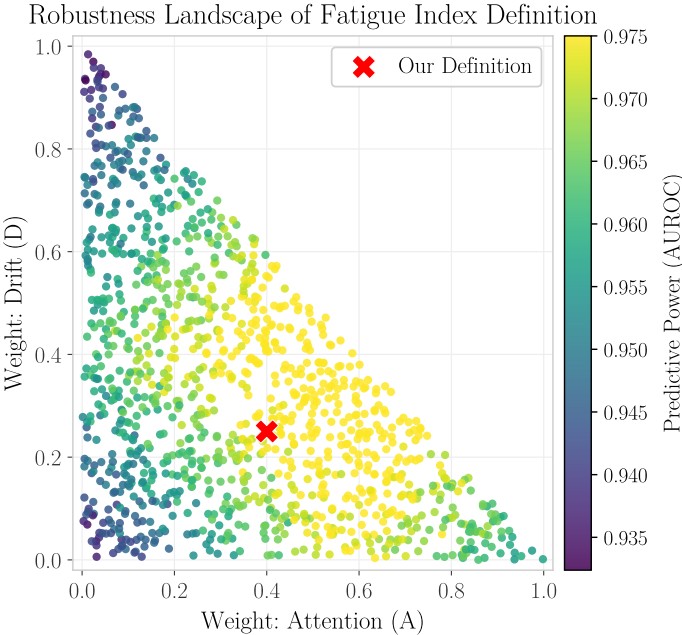

*Figure 8.* Robustness landscape of the Fatigue Index (FI) definition over the weight simplex (A + D + E = 1). Each point represents a candidate weighting of attention (A), drift (D), and entropy (E), colored by its AUROC against downstream degeneration labels. The landscape is smooth and structured rather than erratic, with a broad high-AUROC region concentrated at moderate-to-high attention weights. Our chosen definition (red ) sits within this high-performance region, confirming that FI's predictive validity is not contingent on a precise weight choice, the metric is robust to reasonable perturbations of its definition.

## D. Additional Precision Stress Results

This appendix reports supplementary precision stress experiments supporting Section 5.2. We compare entropy, attention-to-prompt, and embedding drift trajectories under FP16 and 4-bit NF4 decoding on additional datasets.

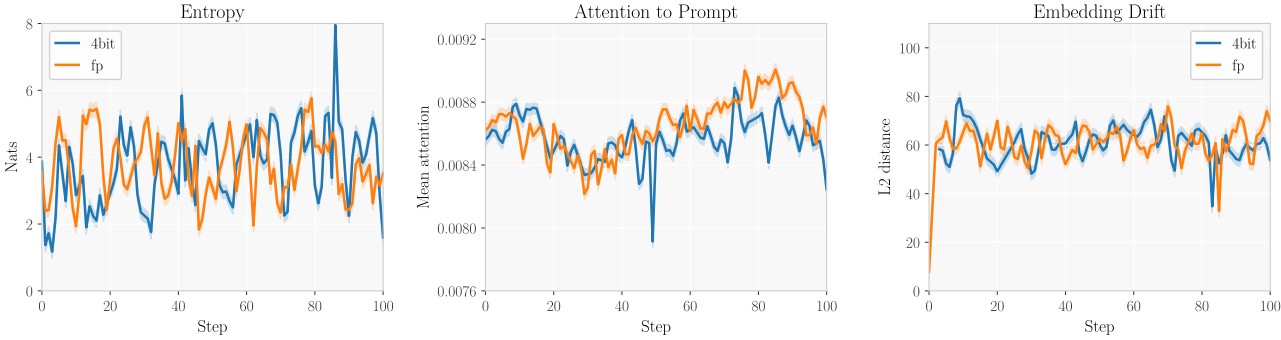

*Figure 9.* Supplementary precision stress results on HotpotQA (FP16 vs. 4-bit NF4, 100 generation steps). Entropy under 4-bit quantization exhibits deeper troughs and higher variance than FP16, consistent with increased susceptibility to predictive calibration collapse. Attention-to-prompt and embedding drift trajectories remain closely matched across precisions, replicating the main finding that quantization selectively disrupts output distribution sharpness while leaving prompt focus and representation stability largely intact.

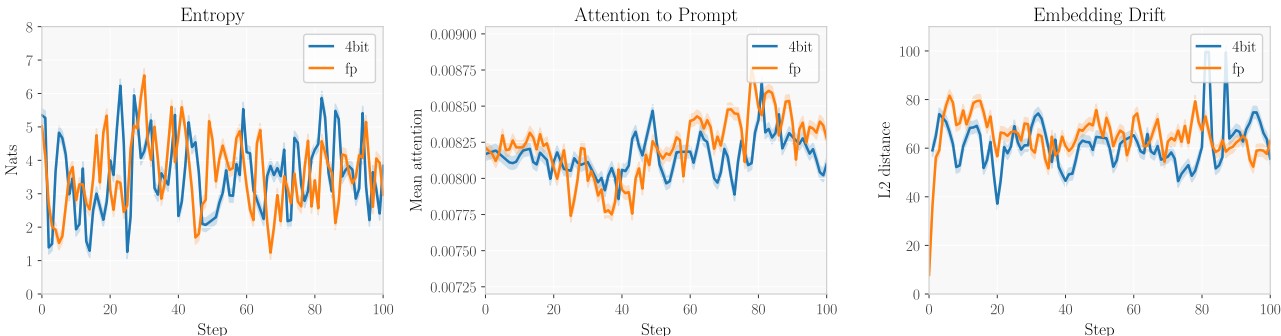

*Figure 10.* Supplementary precision stress results on TriviaQA (FP16 vs. 4-bit NF4, 100 generation steps). Entropy trajectories under 4-bit quantization exhibit greater depth and variability of collapse compared to FP16, replicating the pattern observed on HotpotQA. Attention-to-prompt and embedding drift remain closely matched across precisions, confirming that reduced precision selectively destabilizes predictive calibration rather than prompt focus or representational stability.

## E. Axiomatic Characterization of Additive Fatigue Measures

**Theorem E.1** (Characterization of Additive Fatigue Measures)**.** *Let $F(A_t, E_t, D_t)$ be a fatigue measure satisfying A1–A3 and A5. Then there exist monotone, bounded functions $\phi_A, \phi_E, \phi_D : \mathbb{R} \to [0, 1]$ and weights $w \in \Delta$ such that*

$$FI_t = w_A \, \phi_A(A_t) + w_D \, \phi_D(D_t) + w_E \, \phi_E(E_t),$$

*up to an order-preserving reparameterization.*

This characterization establishes that any additive fatigue measure satisfying the stated axioms necessarily takes the weighted-sum-of-monotone-transforms form. It does not prove that FI recovers a unique causal mechanism of degradation or that the specific weights $(w_A, w_E, w_D)$ are uniquely determined; the theorem constrains the functional form, not the parameter values.

## F. Software and Data

The code and scripts used to compute the Fatigue Index and reproduce all experiments in this paper are publicly available at GitHub Repository. The repository includes implementations of all three inference-time signals (prompt attention, embedding drift, and entropy), the FI aggregation pipeline, and scripts for the stress analyses and scaling experiments. An interactive Colab notebook demonstrating the full FI computation pipeline with visualizations is available at Colab Notebook. All experiments were implemented in PyTorch using the HuggingFace Transformers library. The three datasets used for evaluation, HotpotQA, TriviaQA, and SQuAD, are publicly available under open licenses and are cited in the references. No proprietary data or models were used in this work.

